# Comparison of PSA Values from capillary dried blood spot analysis and venous serum samples

**Lucas Engelage** [1,2,3]*, **Niklas Behnel** [2,4], **Agron Lumiani** [2], **Alexander Tamalunas** [1,5], **Alexander Buchner** [1], **Ronald Sroka** [1,3]

1 Department of Urology, University Hospital, LMU Munich, Munich, Germany, 2 ALTA Klinik, Bielefeld, Germany, 3 Laser-Forschungslabor, LIFE Center, University Hospital, LMU Munich, Munich, Germany, 4 Department of Anaesthesiology and Intensive Care Medicine, Campus Charité Mitte and Campus Virchow-Klinikum, Charité - Universitätsmedizin Berlin, Berlin, Germany, 5 Comprehensive Cancer Center (CCC Munich, LMU), LMU University Hospital Munich, Munich, Germany

* lucas.engelage@altaklinik.de

## Abstract

Prostate-specific antigen (PSA) testing is a key tool in the early detection and monitoring of prostate cancer. While venous blood is the standard matrix for PSA measurement, capillary blood sampling represents a minimally invasive alternative that may facilitate broader access to testing. This prospective study investigated the agreement between capillary and venous measurements of total and free PSA in 224 male patients at ALTA Klinik Bielefeld, Germany. Blood samples were collected between April and June 2021 and analyzed using electrochemiluminescence immunoassays. The results showed strong correlations between capillary and venous samples for both total PSA (r = 0.99, p < 0.01) and free PSA (r = 0.98, p < 0.01). Mean differences were 0.35 ng/mL for total PSA and 0.27 ng/mL for free PSA. Bland-Altman analysis demonstrated good agreement between both methods, with most values falling within the 95% confidence limits. These findings suggest that capillary blood sampling may be a suitable alternative to venous blood for PSA testing, particularly within the clinically relevant range up to 10 ng/mL. Furthermore, the feasibility of determining both total and free PSA in capillary samples allows calculation of the fPSA/tPSA ratio, which may improve diagnostic specificity in early prostate cancer detection.

## Introduction

Prostate cancer is the second most common fatal cancer in men, with 248,000 new cases and more than 34,000 deaths in the United States of America in 2021 [1]. Early detection and monitoring of prostate cancer are often based on the determination of prostate-specific antigen (PSA) concentrations in blood. However, the diagnostic performance of PSA in blood is limited, especially when the concentrations fall in the gray area of 4–10 ng/ml [2]. The use of the ratio of free PSA and total PSA (fPSA/

**Data availability statement:** All relevant data are within the manuscript and its Supporting Information files.

**Funding:** The author(s) received no specific funding for this work.

**Competing interests:** The authors have declared that no competing interests exist.

**Abbreviations:** PSA, Prostate-specific antigen; tPSA, Total prostate-specific antigen; fPSA, Free prostate-specific antigen; DBS, Dried blood spot; ECLIA, Electrochemiluminescence immunoassay; WHO, World Health Organization

tPSA (also known as PSA ratio or %fPSA)) could improve diagnostic accuracy [3,4]. These values are typically measured in blood serum obtained from venipuncture. Alternative methods for PSA determination, including capillary blood sampling from the fingertip, have been investigated [5]. Capillary blood sampling is significantly less invasive and painful than venous puncture. Unlike venous blood collection, capillary methods do not require trained professionals. It can easily be performed by trained patients, making it ideal for use in resource-limited settings or self-testing at home.

Previous studies showed that the determination of PSA from capillary blood is technically feasible [5]. In the present study, all patients gave a venous and a capillary blood sample from the fingertip. In addition, previous studies only investigated the total PSA concentration in dried blood, and the determination of free PSA in dried blood was not performed. Free PSA, which means unbound PSA, is an important factor to determine the probability of a malignant tumor. Using the ratio increases diagnostic specificity and helps distinguish benign from malignant prostate conditions [3,4].

This study investigated the concordance of total and free PSA concentrations between capillary blood and venous serum in 224 patients.

## Materials and methods

### Recruitment

Male patients at the ALTA Klinik (Bielefeld, Germany), requiring PSA concentration determination for diagnostic or therapeutic reasons during the study period were asked to participate. Patients were not pre-selected based on demographic data, diagnoses, symptoms, or medical history. All participants gave written informed consent. The study was approved by the ethics committee of the responsible medical association (Ärztekammer Westfalen-Lippe) and the University of Münster (No. 2020–701-f-S).

### Blood sampling

Two blood samples were obtained from the patients: a capillary blood sample from the fingertip and a venous blood sample from the arm vein according to established protocols. [6–8] Capillary blood from the fingertip was directly added onto a dried blood spot (DBS) card (Ahlstrom-Munksjö, no. 460; Helsinki, Finland). The DBS cards were air-dried at room temperature for at least 4 hours, shielded from direct light, and then shipped to the analysis laboratory without refrigeration.

### Determination of PSA-values

Blood samples were analysed at a dedicated specialized laboratory (MVZ Labor Krone eGbR, Bad Salzuflen, Germany). The DBS cards were first subjected to quality control, after which a 6 mm spot was punched out and transferred into a Deep Well Plate. After elution in 225 µl analysis buffer (Universal Diluent, Roche Diagnostics GmbH, Mannheim, Germany), the samples were incubated for 1 h at 1000 rpm following a short centrifugation (5 min, 4000xg). Venous blood samples were also

centrifuged for 5 min and 4000xg. Electrochemiluminescence Immunoassay (ECLIA) of total PSA and free PSA was performed using standard diagnostic device (Elecsys, Roche Diagnostics GmbH, Mannheim, Germany). The samples were analyzed (Cobas e801, Roche Diagnostics International AG, Rotkreuz, Switzerland) according to the manufacturer's instructions.

### Selection of samples

Patient samples that did not meet the quality control requirements were excluded from processing and analysis. Values below the detection limit (0.014 ng/ml PSA, 0.018 ng/ml free PSA) were excluded from the analysis due to a lack of validity.

A total of 224 male patients were recruited from the ALTA Klinik Bielefeld between April and June 2021. The recruitment and inclusion process is illustrated in Fig 1.

### Statistical analysis

Statistical analyses were performed using the software Python 3.11. The following libraries were used: pandas (for data management), numpy (for numerical calculations), scipy and statsmodels (for statistical tests), matplotlib and seaborn (for visualizing the results). For statistical analysis the correlation according to Bravais-Pearson [9] and the effect size according to Cohen [10] was calculated.

A Bland-Altman diagram [11] was used to analyse systemic differences. To improve its clarity, data points with values above 15 ng/ml were not included in the graph and were analysed separately. This step was taken to ensure a focused presentation of the central data distribution and avoid excessive scatter in the higher ranges, which would have made interpretation more difficult.

## Results

The distribution of total PSA concentrations measured in venous blood samples for the study population (n = 224) is presented in Fig 2.

Total and free PSA values showed strong positive correlations between measurements from capillary blood and blood. The correlation was 0.99 for total PSA and 0.98 for free PSA (p-value: < 0.01). The mean difference was 0.35 ng/ml for total PSA and 0.27 ng/ml for free PSA.

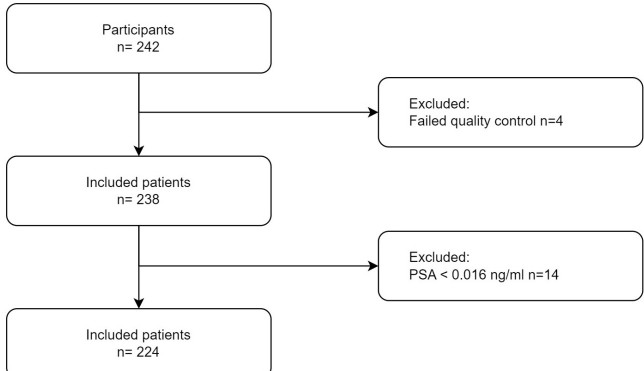

**Fig 1. Overview of recruited participants.** Patients whose PSA values were below technical detection limit were also correctly identified below the detection limit of 0.014 ng/ml (Roche Cobas e801) in 100% of cases (n = 14).

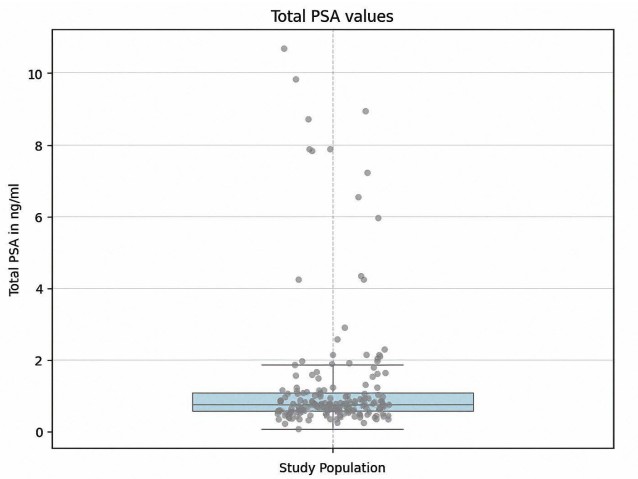

**Fig 2. The boxplot displays the distribution of venous total-PSA values (ng/mL) measured in the study population(n = 224).** The box represents the interquartile range (IQR; 25th–75th percentile), with the horizontal line indicating the median. Individual measurements are overlaid as gray dots. For improved visualization, extreme values above 15 ng/ml (n = 4) were analyzed separately and are not included in this plot.

Due to significant variability and the limited sample size (n = 4), PSA values exceeding 15 ng/ml were excluded from detailed analysis and are discussed within the limitations section.

The Bland-Altman plot displays the differences (capillary – venous) between paired measurements of tPSA against their mean values, illustrating the agreement between both sampling methods (see Fig 3). The red dashed line indicates

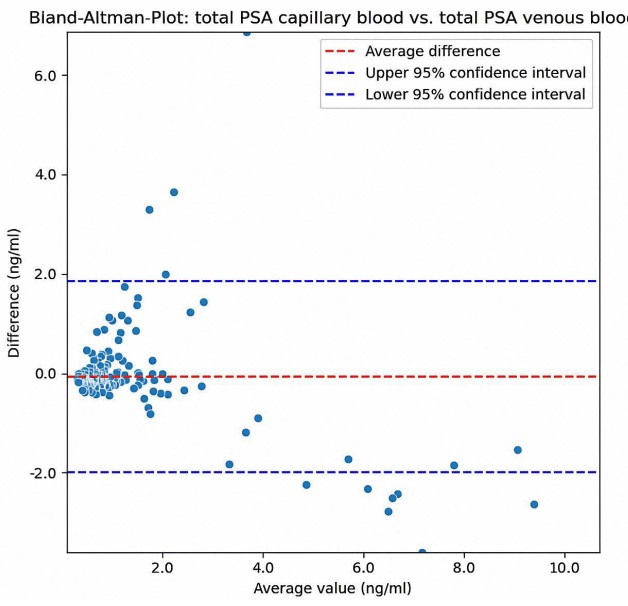

**Fig 3. Bland-Altman plot illustrating the differences (capillary – venous) between paired measurements of tPSA against their average values.** The red dashed line indicates the mean difference, and the blue dashed lines represent the 95% limits of agreement (mean ± 1.96 SD). Higher variability is observed particularly at mean PSA concentrations above 6 ng/ml. Note that values above 15 ng/ml (n = 4) were analysed separately and are not shown in this plot.

the mean difference between the two measurement methods. This line was close to zero, indicating no systematic bias between methods.

The data points were distributed around the mean difference with greater variability at higher mean values (> 6 ng/ml). The blue dashed lines represent the upper and lower 95% confidence limits. Most data points were within these limits, indicating acceptable agreement. Overall, the plot shows a good agreement between the two measurement methods, with some variability.

In comparison, the Bland-Altman plot for free PSA (Fig 4) also showed a mean difference close to zero, indicating no significant systematic bias. The differences were more concentrated at lower mean values, and there was less variability compared to the total PSA in the Fig 3. Most points were within the confidence limits, and the overall dispersion was lower than that in Fig 3.

## Discussion

This study investigated the correlation between the concentrations of total PSA and free PSA measured in capillary blood and venous blood. The results showed a highly significant correlation, suggesting that capillary blood may be a suitable alternative to venous blood sampling for PSA determination.

In the present study, largely consistent results were obtained between the two measurement methods, indicating a good agreement in the determination of either total or free PSA values.

This study used a prospective design with real capillary blood collection from patients in clinical settings, allowing for direct observation of self-sampling under practical conditions. In contrast, some earlier studies, such as van den Brink et al. [12], assessed self-sampling using a topper device, while Wu et al. [13] evaluated gold-based immunochromatographic strips. These approaches differ in sampling methods and settings. By using capillary blood under routine clinical conditions, this study adds complementary data to the existing literature and helps to contextualize the feasibility of self-collection across varying real-world scenarios.

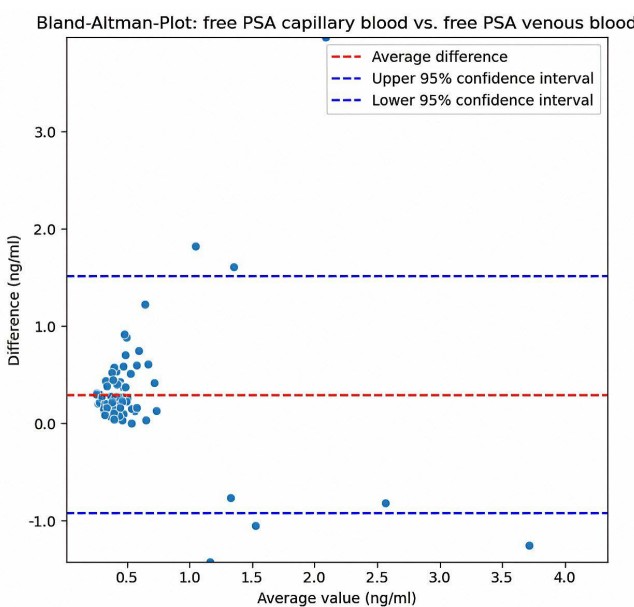

**Fig 4. Bland-Altman plot illustrating the differences (capillary – venous) between paired measurements of free PSA against their mean values.** The red dashed line represents the mean difference, while the blue dashed lines indicate the 95% limits of agreement (mean ± 1.96 SD). Variability was predominantly observed at lower concentrations, with most measurements within the limits of agreement.

The methodological approach in PSA determination across studies warrant careful consideration. This study utilized ECLIA on the Roche Cobas platform, whereas others have employed various analytical approaches such as gold immunochromatographic assays [13] or capillary blood collection with topper devices [12]. Despite using different methodologies, both comparative studies achieved strong correlations with conventional venous sampling – van den Brink et al. reported a correlation coefficient of 0.998 [12], while Wu et al. demonstrated a correlation of 0.960 with their rapid quantitative test [13]. The results of the presented study are consistent with this assessment through the differential determination of free and total PSA concentrations furthering diagnostic reliability by using the fPSA/tPSA ratio [3,4].

Given the general discussion about rigid limit values [14] in the context of diagnostic-therapeutic relevance, mean deviations of 0.35 ng/ml for PSA-values of up to 10 ng/ml can be regarded as acceptable [15]. Although greater variability (up to approximately 30%) was observed in higher PSA ranges (>10 ng/ml), the number of cases was very limited (n = 4), restricting reliable conclusions. Such deviations align with known methodological and biological variabilities of PSA testing, and are clinically less relevant since elevated PSA-values results typically require confirmatory venous measurements and further diagnostic evaluation [15,16]. Therefore, deviations in the higher range are of limited clinical relevance, as the primary function of the initial PSA screening – identifying patients requiring further evaluation – remains preserved.

Analysing the Bland-Altman diagrams, there is slight non-systematic variability between the methods. This is in line with a fundamental problem, as the determination of the PSA value is still a subject to high, partly inherent variability. In addition to the significant biological variability [17,18] of PSA concentrations, considerable variability among different assays has been documented [13,19–22]. Despite efforts at standardization, such as the WHO calibration [23], only harmonization, but not complete standardization [24], has been achieved to date. Deviations in the range of 15–20% are still described [21,24]. The deviations from the gold standard, observed in this study, are therefore within the typical range. Guideline recommendations that call for testing under comparable conditions in the same laboratory [24–26] are difficult to implement, particularly in the context of screening-based preventive examinations, and stand in the way of low-threshold accessibility to such services. Therefore, the lack of standardization remains a problem that needs to be resolved. Early diagnosis is particularly important in the treatment of prostate cancer, which lacks early pathognomonic symptoms [27]. Early diagnoses and broader treatment options can be achieved by improving the availability of low threshold testing and diagnostic options. As the diagnostic value of PSA determination is increased through regular measurement and monitoring [19,27–29], a positive diagnostic effect of the expansion of existing services can be assumed. Capillary blood sampling from the fingertips offers this possibility. A significant advantage is the elimination of incorrect punctures, such as those in patients with difficult venous conditions. These are particularly prevalent in older patients; therefore, puncture is associated with increased effort and pain. Low-invasive capillary sampling is generally considered less invasive and can therefore increase patient acceptance [30].

One possible advantage of capillary blood collection from the fingertips is resource efficiency. While venous blood sampling typically requires professional healthcare personnel, capillary blood sampling can easily be performed by minimally trained individuals or by the patients themselves.

It is important to mention that this study has some limitations. In this study, capillary blood samples were collected by trained medical professionals. This means that pre-analytical errors, which are possible when capillary blood samples are taken by the patients themselves, were not recorded. Therefore, no conclusive statements can be made regarding its use outside the medical environment.

Although the present study design did not allow direct assessment of retest reliability, the strong correlation and minimal mean difference suggest that repeated capillary PSA measurements could yield consistent results. Previous research has indicated intra-individual PSA variability of approximately 10–20% [18]. Current guidelines therefore advocate for confirmatory PSA testing prior to invasive interventions, especially within a diagnostic grey zone. The German S3 Guideline similarly emphasizes informed patient-decision making, recommending initial PSA screening combined with digital rectal examination from the age of 45 years and advocating biopsy only after confirmatory testing or significant PSA velocity. [31]. Further studies should explicitly assess test-retest reliability of the DBS method.

However, further studies involving larger patient populations and the identification of possible pre-analytical sources of error, particularly concerning the sample collection technique and storage, are required before a general recommendation can be made.

## Conclusions

A significant correlation between capillary and venous blood samples for the measurement of total and free PSA was shown, indicating that capillary testing could represent a suitable alternative to venous blood sampling. The analysis of free PSA from DBS samples is a promising approach to improve the diagnostic specificity of the PSA value. In general, the suitability of capillary testing opens a lot of possibilities for improving diagnostic and therapeutic services, above all the possibility of self-sampling.

## Supporting information

**S1 Fig. Supplemental Figure 1. Dataset containing all relevant study data (CSV file).**
(CSV)

## Author contributions

**Conceptualization:** Lucas Engelage.

**Data curation:** Lucas Engelage.

**Formal analysis:** Lucas Engelage.

**Investigation:** Lucas Engelage, Alexander Tamalunas, Alexander Buchner.

**Methodology:** Niklas Behnel.

**Project administration:** Agron Lumiani.

**Supervision:** Agron Lumiani, Ronald Sroka.

**Validation:** Lucas Engelage, Niklas Behnel, Alexander Tamalunas, Alexander Buchner, Ronald Sroka.

**Visualization:** Lucas Engelage, Niklas Behnel.

**Writing – original draft:** Lucas Engelage, Niklas Behnel, Alexander Tamalunas, Alexander Buchner, Ronald Sroka.

**Writing – review & editing:** Lucas Engelage, Niklas Behnel, Alexander Tamalunas, Alexander Buchner, Ronald Sroka.

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
