## [Decision Letter · Decision Letter 0]

3 Sep 2025

Dear Dr. Engelage,

Thank you for submitting your manuscript to PLOS ONE. After careful consideration, we feel that it has merit but does not fully meet PLOS ONE’s publication criteria as it currently stands. Therefore, we invite you to submit a revised version of the manuscript that addresses the points raised during the review process.

We look forward to receiving your revised manuscript.

Kind regards,

Rajeevan Selvaratnam

Academic Editor

PLOS ONE

Journal Requirements:

Reviewers' comments:

Reviewer's Responses to Questions

**Comments to the Author**

1. Is the manuscript technically sound, and do the data support the conclusions?

Reviewer #1: Yes

Reviewer #2: Yes

2. Has the statistical analysis been performed appropriately and rigorously?

Reviewer #1: Yes

Reviewer #2: Yes

3. Have the authors made all data underlying the findings in their manuscript fully available?

Reviewer #1: Yes

Reviewer #2: Yes

4. Is the manuscript presented in an intelligible fashion and written in standard English?

Reviewer #1: Yes

Reviewer #2: Yes

Reviewer #1: The article Comparison of PSA Values from Capillary Dried Blood Spot Analysis and Venous Serum Samples is well written and clearly written, with a well-described methodology and clear conclusions. It describes the use of a new methodology for PSA analysis from capillary blood samples. This is a prospective study conducted on a group of 224 male patients. The primary data, only of total PSA, are available. The discussion is factual and the authors discuss the advantages of capillary blood samples but also mention the limitations of the method and study.

I have only one comment on the Conclusion.

I disagree with the authors' conclusion that The analysis of free PSA from DBS samples is a promising approach to improve the diagnostic specificity of the PSA value.

The authors correctly state in the discussion the general and known problems in determining PSA. It is generally known that freePSA increases the sensitivity and specificity of the determination of total PSA alone. However, in line 250 the authors state that Although the present study design did not allow direct assessment of retest reliability...

It is therefore not possible to objectively evaluate the analytical performance of both methods used. From personal experience, I would assume that the analytical performance of the established and long-standing Roche ECLIA method will be better than methods based on capillary dried blood spots. (But I could be wrong.) In any case, without analytical performance, the claim of improved PSA specificity when using free PSA specifically from DBS samples is irrelevant.

After resolving my comment and minor revision, I recommend the article for publication.

Reviewer #2: Dear Authors,

Your study strongly demonstrates the feasibility of capillary blood samples for PSA testing. The ability to calculate the fPSA/tPSA ratio using DBS is particularly valuable for its potential to increase diagnostic specificity. However, the study requires some minor revisions. These include:

1. Capillary samples were collected by healthcare personnel, so the true "self-sampling" scenario was not evaluated. This should be more clearly emphasized as a limitation.

2. High PSA values (>15 ng/ml) were excluded from the study; the clinical impact of this condition should be further investigated.

It is recommended that these limitations be addressed more clearly and the Discussion section be strengthened accordingly.

**Do you want your identity to be public for this peer review?** For information about this choice, including consent withdrawal, please see our Privacy Policy

Reviewer #1: No

Reviewer #2: No

---

## [Author Response · Author response to Decision Letter 1]

6 Oct 2025

A detailed point-by-point response to all reviewer and editor comments has been uploaded as a separate file titled “PLOS ONE – Response to Reviewers.docx”.

We thank the reviewers and the editor for their constructive feedback and helpful suggestions, which have improved the clarity and quality of our manuscript.

---

## [Editor Report · Decision Letter 1]

12 Oct 2025

Comparison of PSA Values from Capillary Dried Blood Spot Analysis and Venous Serum Samples

PONE-D-25-33962R1

Dear Dr. Engelage,

We’re pleased to inform you that your manuscript has been judged scientifically suitable for publication and will be formally accepted for publication once it meets all outstanding technical requirements.

Kind regards,

Rajeevan Selvaratnam

Academic Editor

PLOS ONE
---

## [Editor Report · Acceptance letter]

PONE-D-25-33962R1

PLOS ONE

Dear Dr. Engelage,

I'm pleased to inform you that your manuscript has been deemed suitable for publication in PLOS ONE. Congratulations! Your manuscript is now being handed over to our production team.

Kind regards,

on behalf of

Dr. Rajeevan Selvaratnam

Academic Editor

PLOS ONE